# Nutritional status, symptom burden, and predictive validity of the Pt-Global web tool/ PG-SGA in CKD patients: A hospital based cross sectional study

Ishfaq Rashid[1], Pramil Tiwari[1]*, Sanjay D'Cruz[2], Shivani Jaswal[3]

**1** Department of Pharmacy Practice, National Institute of Pharmaceutical Education and Research (NIPER), S.A.S. Nagar, Punjab, India, **2** Department of General Medicine, Government Medical College and Hospital (GMCH), Chandigarh, India, **3** Department of Biochemistry, Government Medical College and Hospital (GMCH), Chandigarh, India

* ptiwari@niper.ac.in

## Abstract

**Data Availability Statement:** The data file has been added as S1 Data.

### Background

Despite not being frequently recognized, malnutrition, a consequence of chronic kidney disease, negatively affects morbidity, mortality, functional activity, and patient's quality of life. Management of this condition is made more difficult by the dearth of knowledge regarding the symptom burden brought on by inadequate nutritional status. Additionally, there are multiple tools to evaluate nutritional status in CKD; but, Pt-Global web tool/PG-SGA used in oncology, has not been investigated in chronic kidney disease patients. This study aimed to explore the nutritional status, symptom burden and also investigate the predictive validity of Pt-Global web tool/PG-SGA among pre-dialysis diabetic and non-diabetic chronic kidney disease patients.

### Methodology

This cross-sectional study was carried out at a renal clinic of a tertiary care public teaching hospital. Nutritional status and symptom burden was evaluated by employing a 'Pt-Global web tool/PG-SGA' which is considered as a preeminent interdisciplinary tool in oncology and other chronic catabolic conditions. The predictive validity of the Pt-Global web tool/PG-SGA, referred as overall score for malnutrition was ascertained using Receiver Operating Curves (ROC). The conclusions were drawn using descriptive statistics, correlation, and regression analysis.

### Results

In a sample of 450 pre-dialysis CKD patients, the malnutrition was present in 292(64.9%) patients. Diabetic CKD patients exhibit higher proportion of malnutrition 159(35.3%). The prevalence of malnutrition was exacerbated by eGFR reduction. The overall Pt-Global web tool/PGA-SGA score was significantly influenced by the symptoms of fatigue (81.5%),

**Funding:** The authors received no specific funding for this work.

**Competing interests:** The authors have declared that no competing interests exist.

appetite loss (54.8%), physical pain (45.3%), constipation (31.78%), dry mouth (26.2%), and feeling full quickly (25.8%). The ROC analysis showed that the AUC for the total PG-SGA score was 0.988 (95% CI: 0.976–1.000), indicating that it is a reliable indicator of malnutrition. The sensitivity (84.2%) for identifying malnutrition was low when using the conventional tool cut off score of ≥9. Instead, it was discovered that a score of ≥3 had a greater sensitivity (99.3%) and specificity (44.3%) and was therefore recommended.

## Conclusions

This study not only presents empirical evidence of poor nutritional status in CKD patients but also reveals that it is worse in patients with diabetes, hypoalbuminemia, and poorer kidney function (well recognized risk factors for cardiovascular disease). Early diagnosis and management of symptoms contributing malnutrition will reduce mortality and CKD progression. The Pt-Global web tool/PG-SGA total score of 3 or more appears to be the ideal cut off score for identifying malnutrition, which can be utilized by dietician for improving malnutrition.

## Introduction

Malnutrition describes a state resulting from lack/inadequate intake or uptake of nutrition that leads to altered body composition (decreased fat free mass) and body cell mass [1]. Even though it is not frequently recognized, this complication of chronic kidney disease has been shown to have an impact on prognosis, physical and mental function, and quality of life [2,3]. Additionally, mortality and the fast advancement of CKD to end-stage renal disease are linked to it [3–5].

According to data from the Global Burden of Disease, chronic kidney disease (CKD) ranks among the leading causes of death globally [6]. Earlier, it was predicted that 1.4 million deaths will be attributable to CKD in 2019, a 20% increase from data from 2010 [7]. Low- and middle-income countries have been severely impacted by its disproportionate rise in prevalence and mortality [8,9]. Poor nutritional status or malnutrition is an under-recognized and undertreated condition that has a high clinical impact on patients across different healthcare settings [10–12]. Patients with chronic renal disease frequently have this complication [13,14]. According to the available research, there is a clear connection between poor nutritional status and morbidity, the geriatric population, and gender [15,16]. The effects are catastrophic, especially in terms of substantial clinical and financial costs, lowered immunity, frequent infections, death, general physical and mental deterioration, and poor quality of life [2,3,17,18].

Nutrition shares a strong bond with the renal outcomes. India has reported a huge prevalence of malnutrition (56.7%) among chronic kidney disease patients [13]. The prevalence of malnutrition, however, varies depending on the stages of CKD, gender, and dialysis status [19]. In developing nations, it has been reported that quantification of malnutrition with respect to dialysis status, CKD stages and gender is poorly characterized before the initiation of dialysis [13]. As a result, routine screening with a validated method to determine the patient's current nutritional status is a crucial step in identifying individuals at risk of malnutrition and are consequently more vulnerable to illnesses.

Various nutritional assessment tools in CKD patients have been utilized in clinical practice [20–22]. However, the subjective global assessment (SGA) instrument has been recommended

by the National Kidney Foundation's Kidney Disease/Dialysis Outcomes and Quality Initiative (KDOQI) as the diagnostic benchmark for the nutritional assessment [23]. This frequently used tool (SGA) in clinical practice has undergone significant advances in recent years [24]. This tool has been made now available as a web-based programme (Pt-Global web tool/ PG-SGA) [25] which is practical, affordable, simple to use, and is non-intrusive in character. This tool also offers automatic solutions to the problems after nutritional assessment.

The predictive validity of manual SGA score system regarding the mortality has been already established in conservatively treated chronic kidney disease patients and/or hemodialysis patients [26–28]. However, the predictive validity of Pt-Global web tool/PG-SGA in non-dialysis chronic kidney disease patients has not been investigated so far. Further, because the patient's symptom burden has not yet been clarified, it is questionable what underlying factors exist and contribute to malnutrition [29]. Therefore, it is relevant for a healthcare professional to strive for the early detection of malnutrition among CKD patients through the necessary assessment of nutritional status; and, plan for management of the condition through dietary plans, nutritional goals, and intervention measures.

Given these gaps, this prospective observational study aimed to evaluate the nutritional status, symptom burden, and also investigate the predictive validity of Pt-Global web tool/ PG-SGA among pre-dialysis diabetic and non-diabetic chronic kidney disease patients at a tertiary care public teaching hospital.

## Materials and methods

### Study design, setting and subjects

This prospective cross-sectional study was carried out in the renal clinic, Department of General Medicine at a tertiary care public teaching hospital which receive referrals of chronic kidney disease patients from within and outside the state of location.

Patients with non-dialysis chronic kidney disease of either gender were recruited by convenience sampling from the outpatient department (OPD) between August 2019 and November 2021. The inclusion criteria covered patients ≥18 years old, conscious and alert, non-dialyzed, willing to participate and if their medical file contained all the demographic, clinical and biochemical data required for the study. Patients who were on dialysis, seriously ill, or unable to communicate verbally or mentally were not included in the study.

### Study procedure

The study procedure involved the data collection and patient interview for nutritional status using Pt-Global web too/PG-SGA. Data were gathered using a self-administered case record form that asked for socio-demographic details like age, gender, residency, height (measured with an inch tape), weight (recorded using a standard calibrated weighing scale), body mass index (BMI), social history (smoking/drinking), and socio-economic status (occupation, education, and income) by modified kuppuswamy scale. This case record form was also used to collect the clinical status of the patients (i.e., the diagnosis, biochemical parameter estimation, current medications and the number of comorbidities by charlson comorbidity index).

The ideal overall metric of kidney function is the glomerular filtration rate (GFR). Normal GFR varies according to age, sex, and body size, and declines with age. GFR was estimated by using the National Kidney Foundation recommended CKD-EPI equation web application. Estimated GFR was used to stage CKD as stage 1 (GFR ≥ 90 ml/min), stage 2 (GFR 60–89ml/ min), stage 3 (GFR = 59–30 ml/min), stage 4 (GFR = 15–29 ml/ min) and stage 5 (GFR < 15ml/min) [30].

## Pt-Global web tool/PG-SGA

There are two sections in this tool [25]. The first section is based on patient grading and contains information about the patient's medical history (weight change, changes in food intake, gastrointestinal symptoms, activities, and functional capacity), while the second section is based on physician grading and includes information about disease status (comorbidities and advanced age), metabolic demand (fever, fever duration, and use of corticosteroids), as well as a physical examination (subcutaneous fat loss and signs of muscle wasting).

The evaluation method of the Pt-Global web tool/PG-SGA is automatic and quick. It provides "nutritional triage recommendations" for inadequate nutritional status after completion based on its scoring, and it also gives the user the option to save the nutritional status report on email for later use. The additive score of the Pt-Global web tool PG-SGA is taken into consideration by nutritional triage recommendations in order to determine the precise nutritional interventions. Optimal symptom management is a component of the first line nutritional intervention. The nutritional interventions include patient & family education, symptom management including pharmacologic intervention, and appropriate nutrient intervention (food, nutritional supplements, enteral, or parenteral triage). Nutritional triage is determined by the Pt-Global web tool/PG-SGA score. Scores of 0 or 1 indicate that no immediate action is required, but that treatment-related decisions will be periodically and routinely reviewed; Score 2–3: Pharmacologic intervention as indicated by the symptom survey parameter and pertinent lab values, together with patient and family education provided by a dietitian, nurse, or other clinician; Scores of 4 to 8 indicate the need for dietician intervention in conjunction with medical or nursing care as indicated by symptoms, while a score of $\geq 9$ indicates an urgent need for improved symptom management and/or nutrient intervention options. The license to use this tool was purchased and granted by the copyright owner. This Pt-Global web tool is now available at https://pt-global.org/pt-global-app/. On evaluation the patients were either classified as well nourished (category A), moderately malnourished or suspected of being malnourished (category B), or severely malnourished (category C). The patient-generated component and the professional component's scores were added to create the Pt-Global web tool/PG-overall SGA's score, with a higher score indicating more severe malnutrition. Age, stages of chronic illness, and diagnosis all had an impact on the final score. These factors were retrieved from medical records. This instrument has been validated by Faith Ottery and the Hanze University of Applied Sciences in populations with cancer and (fragile) elderly individuals [25].

## Ethics approval

The concerned Human Ethics Committee has approved the sample and data collection and granted permission to access and use the patient's clinical data. The samples collected were assessed by the Department of Biochemistry at the study site. All eligible participants were informed about the study protocol and written, signed consent was obtained from all the participants before their inclusion in the study. This study was conducted in adherence to the Declaration of Helsinki.

## Statistical analysis

Selected demographic and clinical characteristics of the study participants were evaluated using descriptive statistics. The continuous variables were expressed as mean standard deviation, while the categorical variables were expressed as percentages and compared using the chi-square test. After ensuring that the distribution of continuous variables was normally distributed using the Kolmogorov-Smirnov test or by using analysis of variance (ANOVA) when

comparing more groups, the student's t-test for independent samples was used to compare the continuous variables. A univariate spearman's correlation analysis was also conducted using a simple linear correlation, with the exception of categorical variables, for which the chi-square test was applied. A multivariate linear regression analysis and a multinomial logistic regression analysis were carried out to investigate the variables associated to nutritional status and/or SGA score. To determine the sensitivity, specificity, and ideal PG-SGA total cut off score for identifying malnourished non-dialysis patients, the receiver operating characteristics (ROC) curve was used. An area under the curve (AUC) of 0.9–1 indicates the PG-SGA score was an excellent measure for detecting malnutrition; 0.8–0.89 a good test; and 0.70–0.79 a fair test [31]. Significance was fixed at P <0.05. The statistical analyses were performed by using SPSS software (version 25.0; SPSS Inc, Chicago, IL).

## Results

### Patient characteristics

During the course of study, a total of 510 patients were enrolled. The patients with incomplete datasets (n = 30), those who refused to give their consent (n = 15), and those who had Ig nephropathy/AKI (n = 15) were not included in the analysis.

A total of 450 individuals with chronic kidney disease, aged 53.9±14.2 (265 men and 185 women), were eligible for final analysis. [Table 1] Almost two third (61.6%, 277/450) of the sample were aged between 30 and 59 years old; and more than one third (38.4%, 175/450) were aged ≥60 years.

### Prevalence of malnutrition

According to this web based subjective global assessment tool (SGA), severe malnutrition was present in 152(33.8%) patients, 140(31.1%) patients were mildly or moderately malnourished while 158(35.1%) patients were well-nourished. Male patients were more adversely affected with malnutrition (severe 91 (20.2%) & mild/moderate 78(17.3%)) than female patients [Table 1].

CKD with diabetes were more severely affected with malnutrition (mild/moderate 68 (15.1%) & severe 91(20.2%)) as compared to chronic kidney disease patients without diabetes (mild/moderate 72(16.0%) & severe 61 (13.6%)). The nutritional status was also evaluated for different stages of chronic kidney disease. The prevalence of malnutrition increased with decline of renal residual function; from 3.4% (mild/moderate & severe) in CKD stage 1–2 to 16.2% in CKD stage 3a-3b and 45.3% in CKD stages 4–5 [Table 1].

For sociodemographic details, there was a statistically significant difference in the mean of nutritional status categories for the age groups (adult & elderly), diabetic status (CKD with or without diabetes), CKD stages (stage 1–5) as determined by one-way ANOVA (F (1,448) = 16.56, p<0.005), (F (1,448) = 24.96, p<0.005) and (F (5,444) = 12.66, p<0.005) [Table 1].

For biochemical parameters, a statistically significant difference was observed in the mean of nutritional status categories for urea p<0.005, creatinine p = 0.007, phosphorous p = 0.001, eGFR p<0.005, serum albumin p = 0.004, and alkaline phosphatase p = 0.017 [Table 2].

### Univariate spearman's Rho correlations for diabetes and non-diabetes CKD groups

The univariate spearman correlation provided a significant input on correlation of biochemical parameters based on the diabetic status of these patients. The results showed that with decrease in eGFR, serum albumin, and hemoglobin, the SGA score tends to increase

**Table 1. Baseline demographic characteristics of study participants according to nutritional status.**

| Variable | No of participants n = 450 (%) | Well-nourished SGA-A N = 158 (35.1%) | Mild-Moderate SGA-B N = 140 (31.1%) | Malnourished SGA-C N = 152 (33.8%) | p-value |
|---|---|---|---|---|---|
| **Age groups (Years)** | | | | | ≤0.005 |
| Adult (<60) | 277 (61.6) | 117 (26.0) | 81 (18.0) | 79 (17.6) | |
| Elderly (≥60) | 173 (38.4) | 41 (9.1) | 59 (13.1) | 73 (16.2) | |
| **Gender** | | | | | 0.866 |
| Male | 265 (58.9) | 96 (21.3) | 78 (17.3) | 91 (20.2) | |
| Female | 185 (41.1) | 62 (13.8) | 62 (13.8) | 61 (13.6) | |
| **Body Mass Index [24]** | | | | | 0.763 |
| Underweight | 40 (8.9) | 14 (3.1) | 9 (2.0) | 17 (3.8) | |
| Normal | 215 (47.8) | 72 (16.0) | 71 (15.8) | 72 (16.0) | |
| Overweight | 159 (65.3) | 60 (13.3) | 49 (10.9) | 50 (11.1) | |
| Obese | 36 (8.0) | 12 (2.7) | 11 (2.4) | 13 (2.9) | |
| **Socio-economic status** | | | | | 0.301 |
| Lower | 4 (0.9) | 69 (15.3) | 61 (13.6) | 76 (16.9) | |
| Middle | 240 (53.3) | 87 (19.3) | 77 (17.1) | 76 (16.9) | |
| Higher | 206 (45.8) | 2 (0.4) | 2 (0.4) | 0 (0.0) | |
| **Residence** | | | | | 0.802 |
| Rural | 241 (53.6) | 83 (18.4) | 76 (16.9) | 82 (18.2) | |
| Urban | 209 (46.4) | 75 (16.7) | 64 (14.2) | 70 (15.6) | |
| **Smoking** | | | | | 0.661 |
| Yes | 371 (82.4) | 27 (6.0) | 29 (6.4) | 23 (5.1) | |
| No | 79 (17.6) | 131 (29.1) | 111 (24.7) | 129 (28.7) | |
| **Alcohol** | | | | | 0.680 |
| Yes | 196 (43.6) | 71 (15.8) | 53 (11.8) | 72 (16.0) | |
| No | 254 (56.4) | 87 (19.3) | 87 (19.3) | 80 (17.8) | |
| **Diabetes status** | | | | | ≤0.005 |
| Yes | 210 (46.7) | 51 (11.3) | 68 (15.1) | 91 (20.2) | |
| No | 240 (53.3) | 107 (23.8) | 72 (16.0) | 61 (13.6) | |
| **CKD Stages** | | | | | ≤0.005 |
| CKD Stage 1 | 12 (2.7) | 5 (1.1) | 7 (1.6) | 0 (0.0) | |
| CKD Stage 2 | 18 (4.0) | 10 (2.2) | 7 (1.6) | 1 (0.2) | |
| CKD Stage 3a | 34 (7.6) | 19 (4.2) | 11 (2.4) | 4 (0.9) | |
| CKD Stage 3b | 112 (24.9) | 54 (12.0) | 30 (6.7) | 28 (6.2) | |
| CKD Stage 4 | 194 (43.1) | 62 (13.8) | 60 (13.3) | 72 (16.0) | |
| CKD Stage 5 | 80 (17.8) | 8 (1.8) | 25 (5.6) | 47 (10.4) | |

Baseline demographic characteristics. Data presented as number and percentage. Abbreviations: SGA, subjective global assessment; eGFR, estimated glomerular filtration rate; BMI, body mass index; CKD, Chronic Kidney Disease. P value <0.05 taken as significant. eGFR in all patients were estimated by the Chronic Kidney Disease Epidemiology Collaboration (CKD-EPI) formula. BMI range is categorized according to Ikizler et al. [32].

continuously, while with increase in urea, serum creatinine and serum phosphorus the SGA score also increases in both the groups [Table 3].

Additionally, in CKD patients with diabetes, SGA score was correlated with chloride, calcium, and mean corpuscular hemoglobin, but in CKD patients without diabetes, SGA score was correlated with age. The strongest correlations were found for age (years) in the non-

**Table 2.  Baseline biochemical characteristics of study participants according to nutritional status.**

| Variables | Overall Mean ± SD N = 450 | Well-nourished SGA-A N = 158 (35.1%) | Mild-Moderate SGA-B N = 140 (31.1%) | Malnourished SGA-C N = 152 (33.8%) | P-value |
|---|---|---|---|---|---|
| Age (years) | 53.9 ± 14.2 | 48.2 ± 15.0 | 55.8 ± 13.2 | 58.0 ± 12.0 | 0.000 |
| **Biochemical Parameters** | | | | | |
| Sodium (mg/dL) | 139.3 ± 8.1 | 139.3 ± 11.3 | 139.6 ± 5.5 | 138.8 ± 5.9 | 0.717 |
| Potassium (mEq/L) | 5.0 ± 4.8 | 4.6 ± 0.8 | 4.8 ± 0.6 | 5.5 ± 8.3 | 0.291 |
| Chloride | 102.8 ± 8.2 | 103.4 ± 8.8 | 102.9 ± 8.0 | 102.0 ± 8.2 | 0.318 |
| Urea (mg/dL) | 89.2 ± 68.5 | 71.8 ± 50.0 | 89.1 ± 86.2 | 107.5 ± 62.2 | **0.000** |
| Creatinine (mg/dL) | 3.1 ± 3.0 | 2.6 ± 4.2 | 2.9 ± 1.8 | 3.7 ± 3.0 | **0.007** |
| Calcium (mg/dL) | 9.0 ± 1.3 | 9.1 ± 1.1 | 9.0 ± 1.4 | 8.8 ± 1.3 | 0.158 |
| Phosphorus (mg/dL) | 4.6 ± 1.7 | 4.2 ± 1.5 | 4.7 ± 1.6 | 5.0 ± 1.8 | **0.001** |
| Uric Acid (mg/dL) | 7.6 ± 2.1 | 7.3 ± 1.9 | 7.5 ± 2.3 | 7.9 ± 2.1 | 0.083 |
| eGFR (ml/min/1.73m$^2$) | 29.7 ± 19.0 | 36.3 ± 19.2 | 31.3 ± 21.5 | 21.2 ± 12.2 | **0.000** |
| Total protein (g/dL) | 7.1 ± 0.86 | 7.2 ± 0.87 | 7.1 ± 0.84 | 7.0 ± 0.86 | 0.102 |
| S Albumin (g/dL) | 4.0 ± 0.69 | 4.1 ± 0.67 | 4.0 ± 0.73 | 3.8 ± 0.65 | **0.004** |
| ALP (IU/L) | 114.0 ± 49.9 | 105.0 ± 40.9 | 118.3 ± 49.9 | 119.6 ± 56.8 | **0.017** |
| Hemoglobin (g/L) | 10.6 ± 8.8 | 11.3 ± 2.3 | 10.1 ± 2.3 | 10.6 ± 5.2 | 0.079 |
| MCV (fL) | 85.9 ± 8.8 | 85.9 ±6.9 | 86.0 ± 11.1 | 85.9 ± 8.8 | 0.925 |
| MCH (pg) | 27.5 ± 3.4 | 27.6 ± 2.18 | 27.2 ±2.23 | 27.5 ± 3.43 | 0.196 |
| TLC /L | 13.4 ± 42.6 | 15.2 ± 64.0 | 14.2 ± 34.4 | 10.9 ± 6.7 | 0.656 |
| TC (g/L) | 188.9 ± 48.7 | 192.8 ± 48.7 | 190.3 ± 58.1 | 183.5 ± 38.0 | 0.223 |
| TG (g/L) | 169.0 ± 102.6 | 169.3 ± 63.1 | 164.0 ±66.4 | 173.4 ± 151.8 | 0.738 |
| HDL-c (g/L) | 48.2 ± 12.7 | 48.0 ± 13.5 | 49.2 ± 13.2 | 47.5 ± 11.4 | 0.523 |
| LDL-c (g/L) | 103.0 ± 38.3 | 103.9 ± 34.9 | 106.2 ± 46.9 | 99.2 ± 32.3 | 0.280 |
| VLDL (g/L) | 34.1 ± 14.5 | 34.3 ± 11.5 | 33.4 ± 14.5 | 34.4 ± 17.2 | 0.818 |
| **Anthropometry** | | | | | |
| Height (cm) | 163.7 ± 8.8 | 164.3 ± 8.92 | 163.4 ± 8.5 | 163.3 ± 8.97 | 0.527 |
| Weight (kg) | 64.4 ± 13.3 | 65.5 ± 13.2 | 64.4 ± 11.8 | 63.3 ± 14.5 | 0.350 |
| BMI (kg/m$^2$) | 23.9 ± 4.2 | 24.1 ± 4.05 | 24.0 ± 4.01 | 23.6 ± 4.6 | 0.524 |
| SBP (mmHg) | 149.8 ± 24.5 | 148.0 ± 24.2 | 150.3 ± 24.7 | 149.8 ±24.5 | 0.480 |
| DBP (mmHg) | 88.0 ± 13.7 | 89.0 ± 14.5 | 87.4 ± 12.8 | 87.7 ± 13.6 | 0.556 |

Baseline biochemical characteristics. Data presented as mean with standard deviation. Abbreviations: SGA, subjective global assessment; eGFR, estimated glomerular filtration rate; SBP, Systolic blood pressure; DBP, Diastolic blood pressure; BMI, body mass index; ALP, Alkaline Phosphatase; S-Albumin, serum albumin; TC, Total Cholesterol; TG, Triglycerides; TLC, Total Leucocyte count; HDL, High density lipoprotein; LDL, Low density lipoprotein; VLDL, Very low-density lipoprotein; eGFR in all patients were estimated by the Chronic Kidney Disease Epidemiology Collaboration (CKD-EPI) formula.

diabetic CKD group (rho = 0.41; p0.01) and for creatinine in the CKD patients with diabetes (rho = 0.28; p0.01).

## Association of different covariates with Pt-Global web tool/PG-SGA score

A multivariable linear regression was conducted to predict Pt-Global web tool/PG-SGA score from biochemical & demographic parameters like eGFR, phosphorous, glycated hemoglobin (Hb1Ac), serum albumin and age. These variables have statistically significantly predicted the SGA score (F (17, 432) = 8.98, R2 = 0.261, p<0.05). However, the three variables phosphorous (β = 1.728; 95% CI, 0.700 to 0.678; P = 0.039), glycated hemoglobin (HB1Ac) (β = 0.186; 95%

**Table 3. Univariate spearman's Rho correlations of SGA score with other parameters in CKD patients.**

| Variables | CKD with diabetes (N = 210) | CKD without diabetes (N = 240) |
|---|---|---|
| | Rho correlations with SGA>1 | |
| Age (years) | 0.44 | 0.41** |
| Gender (male/female) | 0.07 | -0.05 |
| BMI (kg/m$^2$) | -0.07 | -0.10 |
| Sodium (mg/dL) | -0.14* | -0.15* |
| Potassium (mEq/L) | -0.03 | 0.09 |
| Chloride | -0.20** | 0.03 |
| Urea (mg/dL) | 0.21** | 0.34** |
| Creatinine (mg/dL) | 0.28** | 0.28** |
| Calcium (mg/dL) | -0.16* | -0.10 |
| Phosphorus (mg/dL) | 0.24** | 0.15* |
| Uric Acid (mg/dL) | 0.0 | 0.13 |
| eGFR (ml/min/1.73m$^2$) | -0.27** | -0.39** |
| Albumin (g/dL) | -0.18** | -0.14* |
| Hemoglobin (g/L) | -0.21** | -0.30** |
| MCV (fL) | -0.04 | 0.05 |
| MCH (pg) | -0.20** | 0.0 |
| TC (g/L) | 0.05 | -0.08 |
| TG (g/L) | -0.03 | -0.31 |
| HDL-c (g/L) | 0.02 | 0.02 |
| LDL-c (g/L) | 0.04 | -0.08 |
| VLDL (g/L) | -0.01 | -0.07 |

Significant correlations are marked as

** Correlation is significant at the 0.01 level (2-tailed) and

*Correlation is significant at the 0.05 level (2-tailed).

CI, 0.050 to 0.322; P = 0.007); and age (β = .115; 95% CI, 0.076 to 0.154; P = 0.000) added statistically significantly to the prediction, (p < .05) positively. A positive coefficient designates that the mean of the dependent variable is directly proportional to the value of an independent variable. Furthermore, the variables like eGFR (β = − 0.060; 95% CI, − 0.093 to − 0.027; P = 0.000) and serum albumin (β = − 0.060; 95% CI, − 2.068 to − 0.457; P = 0.002) were negatively associated with the SGA score. Multicollinearity between the independent variables was not observed.

The multiple correlation coefficient (R = 0.551) showed that the dependent variable (SGA score) may be predicted with reasonable accuracy. According to the coefficient of determination (R2 value = 0.261), 26.1% of variance in the dependent variable (SGA score) is explained by these independent variables (biochemical parameters).

A multinomial logistic regression analysis reported that age group (β = 0.959; 95% CI, 1.584 to 4.293; P = 0.000), and diabetes status (β = 1.177; 95% CI, 1.897 to 5.549; P = 0.000) were statistically significantly predicting the nutritional status. The nagelkerke pseudo R$^2$ = 0.110 represents that 11.0% of variance in the nutritional status is explained by the model. Adult age group was found to be more affected with malnutrition, however the box-plot showed that with increase in age, Pt-Global web too/PG-SGA score also tends to increase. The Fig 1 represents the relationship between age and Pt-Global web tool/PG-SGA score.

The regression analysis was also extended to CKD patients based on diabetic status to predict SGA score from biochemical parameters. In non-diabetes CKD patients, the variables

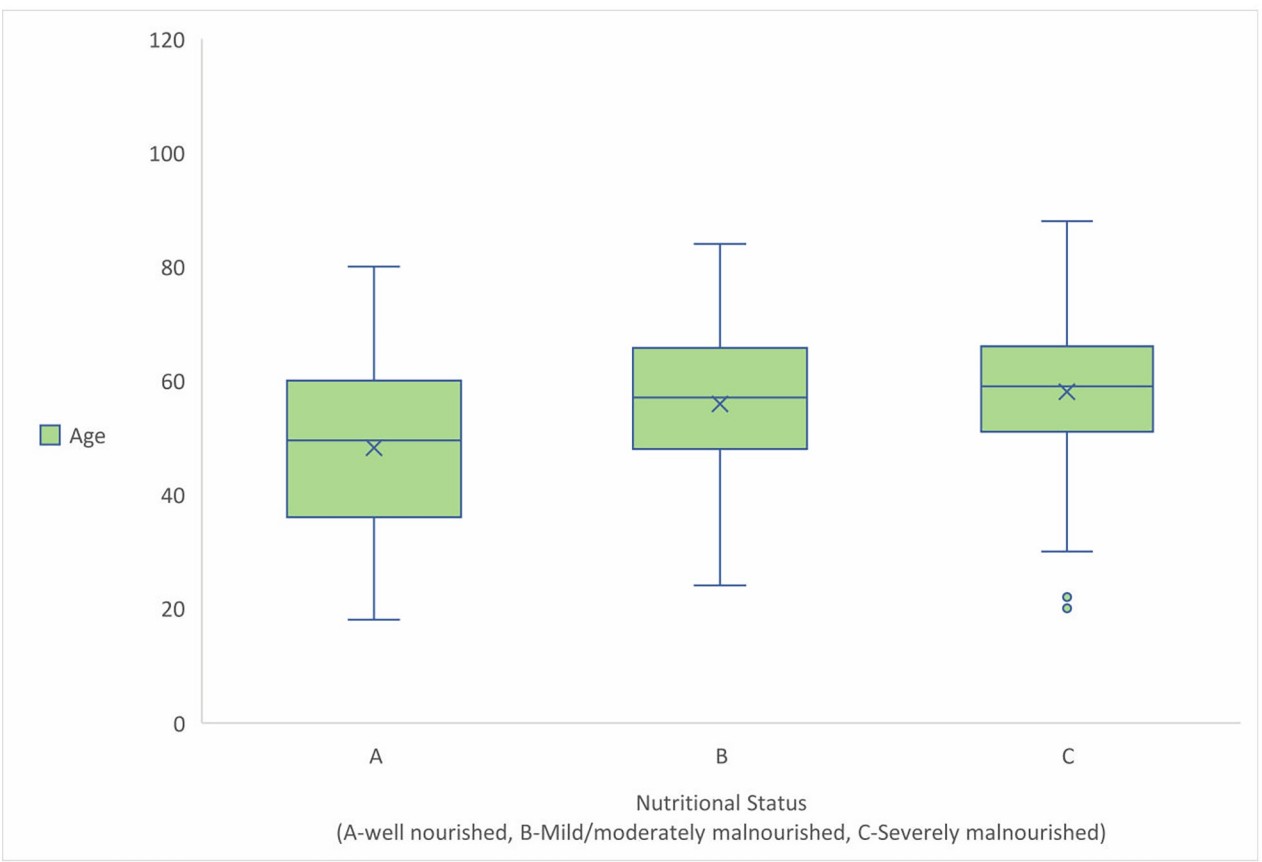

**Fig 1. Relationship between age and nutritional status.**

(urea p = 0.029, hemoglobin p = 0.002 and age p≤0.005) are significantly predicting the dependent variable (SGA score) while in case of CKD patients with diabetes the variables (phosphorous p = 0.02, glycated hemoglobin (Hb1Ac) p = 0.02, and albumin p = 0.05) are statistically significantly predicting the dependent variable (SGA score).

## Determinants of malnutrition

The Pt-Global web tool/PG-SGA parameters (weight, food intake, symptom score, activities & function) also known as PG-SGA short form were evaluated to explain their contribution towards the poor nutritional status. The symptom score parameter was affected in 81.8% patients. The activities and function parameters (79.6%) have also largely contributed to malnutrition followed by changes in food intake (69.6%) and weight change (51.6%). The Fig 2 represents the frequency of individual Pt-Global web tool/PG-SGA parameters towards malnutrition and/or SGA score.

## Symptom burden (symptom score)

This parameter was further evaluated for the problems that have kept the patients away from eating enough prior two weeks of nutritional status evaluation. The results demonstrated that the problems like fatigue (81.5%), loss of appetite (54.8%), body pain (45.3%), constipation (31.78%), dry mouth (26.2%), and feel full quickly (25.8%) were the major contributing factors for higher symptom score in these patients [Table 4].

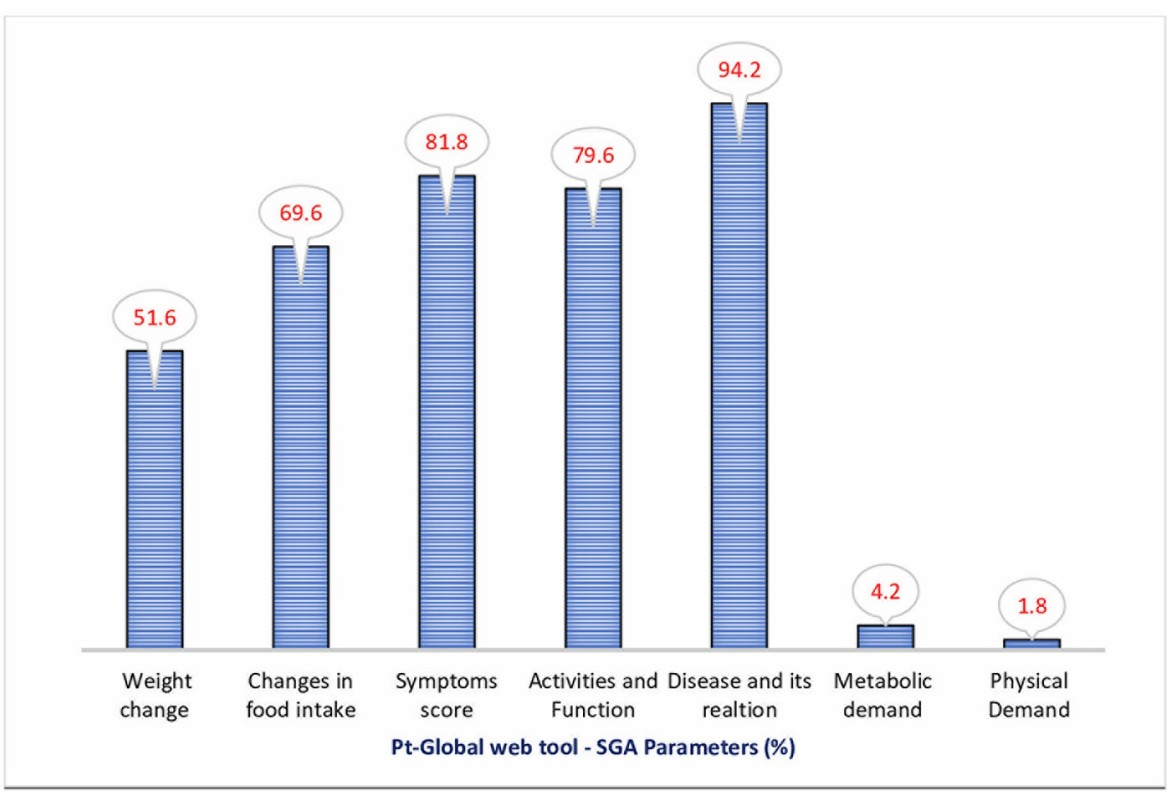

**Fig 2. Frequency of individual Pt-Global web tool/PG-SGA parameters.**

**Table 4. Relationship between malnutrition and nutrition impact symptoms.**

| Variables | No of participants n = 450(%) | Well-nourished SGA-A N = 158 | Malnourished SGA-C N = 292 | p-value |
|---|---|---|---|---|
| PT-Global web tool Score (median) | 11 | 4 | 16 | ≤0.001 |
| **Nutrition impact symptoms** | | | | |
| No appetite, just did not feel like eating | 244(54.8) | 4(0.9) | 240(53.3) | ≤0.001 |
| Nausea | 26(5.8) | 0.0 | 26(5.8) | ≤0.001 |
| Constipation | 143(31.7) | 2(0.4) | 141(31.3) | ≤0.001 |
| Mouth sores | 1(0.2) | 1(0.2) | 0.0 | 0.174 |
| Things taste funny or have no taste | 92(20.4) | 1(0.2) | 91(20.2) | ≤0.001 |
| Problems swallowing | 11(2.4) | 0.0 | 11(2.4) | 0.047 |
| Pain | 204(45.3) | 4(0.9) | 200(44.4) | ≤0.001 |
| Vomiting | 34(7.6) | 0.0 | 34(7.6) | ≤0.001 |
| Diarrhea | 17(3.8) | 0.0 | 17(3.8) | 0.002 |
| Dry Mouth | 118(26.2) | 1(0.2) | 117(26.0) | ≤0.001 |
| Smells bother me | 2(0.4) | 0.0 | 2(0.4) | 0.297 |
| Feel full quickly | 116(25.8) | 4(0.9) | 112(24.9) | ≤0.001 |
| Fatigue | 367(81.5) | 90(20.0) | 277(61.5) | ≤0.001 |
| Other Problems (Depression, Money, dental Problems) | 19(4.2) | 0.0 | 19(4.2) | 0.001 |

A significant mean difference was observed between the nutritional status and symptom burden parameters [Table 4]. This involved the higher proportion of problems like fatigue (61.5%, P≤0.001), loss of appetite (53.3%, P≤0.001), body pain (44.4%, P≤0.001), constipation (31.3%, P≤0.001), dry mouth (26.0%, P≤0.001), and feel full quickly (24.9%, P≤0.001) in patients who are malnourished.

## Mobility functions

The "activities and function parameter" was further examined, and the findings revealed that 'patients with not my normal self but able to be up and about with fairly normal activities' (33.11%), and 'patients not feeling up to most things, but in bed or chair less than half a day' (28.44%) were the major risk factors contributing higher grades/score of decrease in activities and function parameter, which led to malnutrition.

## Predictive validity of Pt-Global web tool/PG-SGA

The findings of the ROC analysis showed that the AUC was 0.988 (95% CI: 0.976–1.000), demonstrating that the overall PG-SGA score was a reliable indicator of malnutrition. The Receiver Operating Characteristics (ROC) plot of the sensitivity and specificity of the Pt-Global web tool/PG-SGA score for predicting malnutrition is shown in Fig 3.

In this patient pool, a PG-SGA total cut off score of ≥9 as per the author of the tool [18] only yielded a sensitivity of 84.2% and specificity of 99.4%. The sensitivity was increased to 99.3% and specificity comes down to 44.3% when the total PG-SGA score was lowered to ≥3. A sensitivity of 99.7% and specificity of 19.6% was observed when Pt-Global web tool/PG-SGA total score was lowered to ≥2.

As per the ROC analysis, the total score cut off ≥3 was considered the most appropriate score to indicate malnutrition and also invites critical need for intervention among non-dialysis CKD patients. The coordinates of the ROC analysis for the total Pt-Global web tool/PG-SGA score (n = 450) are summarized in Table 5.

## Discussion

This prospective study provides a comprehensive overview of the nutritional status (stratified by the diabetic status, CKD stages, and gender) and symptom burden among chronic kidney disease patients before they underwent on dialysis. To the best of the knowledge of researcher's, this is the first time a digital tool "Pt-Global web tool/PG-SGA" has been employed into clinical practice for the evaluation of nutritional status among CKD patients. This study has also investigated the predictive validity of "Pt-Global web tool/PG-SGA" in this particular population. Given the benefits over a manual SGA instrument, this web tool has proven itself as the most preeminent interdisciplinary approach of determining nutritional status.

According to Pt-Global web tool/PG-SGA, our results indicated that 140(31.1%) patients were mild/moderately malnourished; this number was found to be smaller when compared with the results reported by Tan et al. in Chinese patients 2016, 131(44.8%) [33]. The study conducted by Espinosa et al. in Mexican patients reported 1996, 40(44.4%) of prevalence in the same population [34]. The findings of this investigation differed slightly from those of Australian patients 85(40.5%) reported by Chan et al. in 2014 [35].

Further, the results of the current study also showed that 152(33.8%) patients were severely malnourished, which was low as compared to the neighboring states Haryana, India reported by Aggarwal et al. 2018, 58(58.0%) [36], and Uttar Pradesh, India reported by Prakash et al. 2007, 131(64.5%) [37] despite having the same food habits, but the results were twice as high as the results reported by Jagadeswaran et al. 2019, 19(14.7%) Andhra Pradesh, India [38].

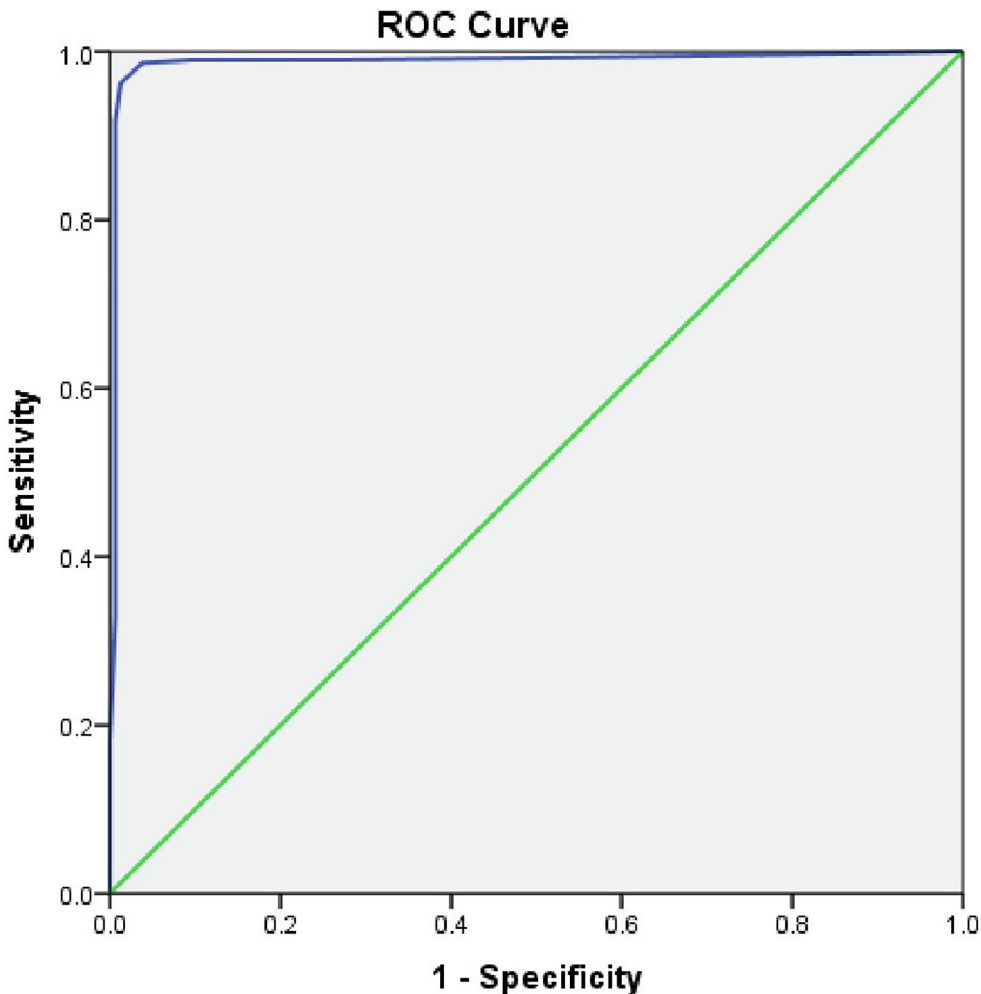

**Fig 3. Receiver Operating Characteristics (ROC) plot of the sensitivity and specificity of the Pt-Global web tool/ PG-SGA score for predicting malnutrition.**

In this study, the overall prevalence of malnutrition (mild/moderate and severe) (64.9%) was found to be higher in comparison to the pooled prevalence 44.2% (range 20.7–70.6%) of malnutrition among non-dialysis chronic kidney disease patients reported in the meta-analysis performed by Rashid et al. [13]. Inflammation, oxidative stress, carbonyl stress, hormonal imbalances, decreased nutrient absorption from an oedematose gut, increased protein loss during dialysis, particularly peritoneal dialysis, and metabolic acidosis are some of the multiple causes of malnutrition in CKD [39–42].

Our findings also provided insights on the nutritional status based on diabetes, CKD stages and gender. Diabetes mellitus is considered as the most common cause of end stage renal disease. The disproportionate increase in the prevalence of diabetic chronic kidney disease or patients with end stage renal disease is narrated as a real epidemic [43] with an appalling prognosis [44]. In addition, diabetic CKD patients with poor nutritional status are also associated with adverse renal outcomes. The current study also evaluated the nutritional status in chronic kidney disease patients based on the diabetic status of participants. The results demonstrate that the patients with diabetes were substantially affected with malnutrition {severely malnourished 91(20.2%)} than non-diabetes CKD group {severely malnourished 61 (13.6%)}.

**Table 5. Coordinates of the receiver operating characteristics (ROC) curve for total Pt-Global web tool/PG-SGA score.**

| Score | Sensitivity | 1—Specificity |
|---|---|---|
| .0000 | 1.000 | 1.000 |
| 1.5000 | 1.000 | 0.987 |
| 2.5000 | 0.997 | 0.804 |
| 3.5000 | 0.993 | 0.557 |
| 4.5000 | 0.990 | 0.266 |
| 5.5000 | 0.990 | 0.095 |
| 6.5000 | 0.986 | 0.038 |
| 7.5000 | 0.962 | 0.013 |
| 8.5000 | 0.918 | 0.006 |
| 9.5000 | 0.842 | 0.006 |
| 10.5000 | 0.777 | 0.006 |
| 11.5000 | 0.705 | 0.006 |
| 12.5000 | 0.651 | 0.006 |
| 13.5000 | 0.596 | 0.006 |
| 14.5000 | 0.551 | 0.006 |
| 15.5000 | 0.517 | 0.006 |
| 16.5000 | 0.473 | 0.006 |
| 17.5000 | 0.332 | 0.006 |
| 18.5000 | 0.182 | 0.000 |
| 19.5000 | 0.099 | 0.000 |
| 20.5000 | 0.027 | 0.000 |
| 21.5000 | 0.007 | 0.000 |
| 22.5000 | 0.003 | 0.000 |
| 24.0000 | 0.000 | 0.000 |

The study of mechanism behind this larger prevalence was outside the scope of the study. However, the existing literature points to the facts that it may be due to poor glycemic control [45], insulin deprivation (the anabolic effects of insulin on protein homeostasis appear to be impaired in patients with type 1 diabetes mellitus) [46], higher resting energy expenditure [47], and restrictive dietary advice [48].

The results of this investigation showed that patients with chronic renal disease exhibit early signs of malnutrition. The prevalence of malnutrition was found to be increased {3.4% (mild/moderate & severe) in CKD stage 1–2 to 16.2% in CKD stage 3a-3b and 45.3% in CKD stages 4–5} with the decline of renal residual function, supplementing the data from previous reports both in pediatric and adult chronic kidney disease patients [49–51] [Table 1]. This pattern was also similar in results reported by Anupama et al. [52] in chronic kidney disease patients. This decrease in renal function potentiates loss of appetite, reduced food intake due to dietary restriction placed on these patients coupled with inadequate nutritional status monitoring, vomiting, diarrhea, hormonal imbalance etc. [53], which contribute significantly to the poor nutritional status in the latter stages of chronic kidney disease. This implies that routine monitoring of nutritional status is imperative at the early stages of chronic kidney disease, as it is more difficult to treat malnutrition when it progresses to higher stages of chronic kidney disease.

Male patients 91 (20.2%) were found to have a greater prevalence of malnutrition than female patients 61 (13.6%). Although the specific cause is not well understood, the observation of an increase in muscle mass loss and protein depletion in male patients suffering from

chronic kidney disease supports this. This might also be due to the fact that men are more likely to seek medical attention than women. Hormonal influences, however, could not be completely ruled out. Tayyem et al., Oluseyi et al., and Stenvinkel et al. also corroborated these findings [54–56].

Adult age group 79(17.6%) was found to be more severely malnourished as compared to elderly age group 73(16.2%). This pattern is surprising because aging is significantly associated with malnutrition in the elderly even without chronic kidney disease [15]. However, the box plot between age and nutritional status showed with increase in age, SGA score continues to increase thus poor nutritional status [Fig 2].

Among non-diabetes CKD patients, the multiple regression analysis showed that the biochemical parameters (urea p = 0.029, hemoglobin p = 0.002 and age p≤0.005) are significantly predicting the SGA score, while in diabetic CKD group the biochemical parameters (phosphorous p = 0.02, glycated hemoglobin (Hb1Ac) p = 0.02, and albumin p = 0.05) were found to be statistically significantly predicting the SGA score. Diabetic chronic kidney disease, end stage renal disease and hypoalbuminemia are well recognized risk factors for cardiovascular disease [57,58].

The evaluation of symptom burden and the predictive validity of the Pt-Global web tool/ PG-SGA comprised the study's main finding. This web application provided a clear grasp of the factors that make up the symptom score. The results of the ROC analysis revealed that the overall PG-SGA score was a credible indication of malnutrition, with an AUC of 0.988 (95% CI: 0.976–1.000).

The importance of understanding the symptom burden is amply illustrated in the current study, where malnourished patients account for a staggering 80.3% of the symptom score while well-nourished patients make up 19.7%. Almutary et al [59] have also highlighted the significant symptom burden among dialysis patients. Notably the symptoms like fatigue, loss of appetite, pain anywhere in the body, constipation, dry mouth, feel full quickly had a strong impact on Pt-Global web tool/PG-SGA total score. This suggests that addressing malnutrition requires multifaceted approach that address both the social determinants of health to improve access to affordable food, as well as treatment of symptoms and underlying potentially medical issues that limit intake. A detailed understanding of symptom clusters may contribute to the quality of life and treatment priorities [60].

## Conclusions

According to this study's findings, around six out of ten non-dialysis CKD patients experienced malnutrition. Furthermore, it has been demonstrated that individuals with diabetes, hypoalbuminemia, and decreased renal function experience greater malnutrition. The results also revealed that the optimal cutoff threshold for diagnosing malnutrition on the Pt-Global web tool/PG-SGA appears to be a total score of ≥3. The symptoms of fatigue, loss of appetite, body pain, constipation, dry mouth, and feeling full quickly substantially exacerbated the malnutrition. These findings will encourage the usage of the P Pt-Global web tool/PG-SGA for nutritional status evaluation. Early emphasis on nutrition in chronic kidney disease patients holds the key for better health outcomes which sequentially can retard the CKD progression and increase the longevity of these patients.

## Supporting information

**S1 Data. Contains the raw data.**
(XLSX)

## Acknowledgments

The authors would like to extend their gratitude towards all the stakeholders who particiapted in the present study and those who carried out the extensive clinical and laboratory work at the clinical site. The authors also ackbowledge the generous support from the mentor institute.

## Author Contributions

**Conceptualization:** Ishfaq Rashid, Pramil Tiwari, Sanjay D'Cruz, Shivani Jaswal.

**Data curation:** Ishfaq Rashid.

**Formal analysis:** Ishfaq Rashid.

**Funding acquisition:** Ishfaq Rashid.

**Investigation:** Ishfaq Rashid.

**Methodology:** Ishfaq Rashid.

**Project administration:** Pramil Tiwari, Sanjay D'Cruz, Shivani Jaswal.

**Resources:** Pramil Tiwari.

**Supervision:** Pramil Tiwari, Sanjay D'Cruz, Shivani Jaswal.

**Visualization:** Pramil Tiwari, Sanjay D'Cruz, Shivani Jaswal.

**Writing – original draft:** Ishfaq Rashid, Pramil Tiwari.

**Writing – review & editing:** Ishfaq Rashid, Pramil Tiwari, Sanjay D'Cruz.

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
