## [Decision Letter · Decision Letter 0]

26 Sep 2022

PGPH-D-22-01367

Nutritional status, symptom burden, and predictive validity of the Pt-Global web tool/PG-SGA in CKD patients: a hospital based cross sectional study.

Dear Dr. Pramil Tiwari,

Thank you for submitting your manuscript to PLOS Global Public Health. After careful consideration, we feel that it has merit but does not fully meet PLOS Global Public Health’s publication criteria as it currently stands. Therefore, we invite you to submit a revised version of the manuscript that addresses the points raised during the review process.

We look forward to receiving your revised manuscript.

Kind regards,

Ransome Eke, M.D., Ph.D., MPH

Academic Editor

Journal Requirements:

2.In the online submission form, you indicated that "Data shall be produced upon request and in accordance with the ethical standards." All PLOS journals now require all data underlying the findings described in their manuscript to be freely available to other researchers, either 1. In a public repository, 2. Within the manuscript itself, or 3. Uploaded as supplementary information.

Additional Editor Comments (if provided):

Please address the concerns of the reviewers below.

Reviewers' comments:

Reviewer's Responses to Questions

**Comments to the Author**

1. Does this manuscript meet PLOS Global Public Health’s publication criteria? Is the manuscript technically sound, and do the data support the conclusions? The manuscript must describe methodologically and ethically rigorous research with conclusions that are appropriately drawn based on the data presented.

Reviewer #1: Yes

Reviewer #2: Partly

2. Has the statistical analysis been performed appropriately and rigorously?

Reviewer #1: Yes

Reviewer #2: N/A

3. Have the authors made all data underlying the findings in their manuscript fully available (please refer to the Data Availability Statement at the start of the manuscript PDF file)?

Reviewer #1: Yes

Reviewer #2: Yes

4. Is the manuscript presented in an intelligible fashion and written in standard English?

Reviewer #1: Yes

Reviewer #2: No

5. Review Comments to the Author

Reviewer #1: Dear author(s),

Thank you very much for your high-quality paper on investigating the validity of the nutritional assessment tools and the relevant outcomes for chronic kidney disease patients.

I have the following only two recommendations for your manuscript:

1. line 87: clarify the sentence - "...problems after nutritional." or "...problems after nutritional assessment."

2. Tables 1 and 4: standardized to 1 d.p. for those zero results.

Reviewer #2: 1. The study is not well organized; not understandable for the readers.

2. The objectives of the study are not well settled.

3. The study has many outcomes/objectives as indicated indifferent parts/segments of it

4. The gap of the study is also not well dressed and not well focused

5. Study design is not clear/lack of consistency and appropriateness

6. The analysis part need modification (when to use and report p- value, association report writing and table association)

7. The study finds also not clearly stated (not understandable)

8. Since the above things are there, reviewing on discussion part is not done

9. Conclusion and Recommendation should be done on the study finds only in the precise form.

10. Same of the references are not update

6. PLOS authors have the option to publish the peer review history of their article (what does this mean?). If published, this will include your full peer review and any attached files.

**Do you want your identity to be public for this peer review?** For information about this choice, including consent withdrawal, please see our Privacy Policy.

Reviewer #1: **Yes: **Yuen Ling LEUNG

Reviewer #2: No

---

## [Decision Letter · Decision Letter 1]

15 Nov 2022

PGPH-D-22-01367R1

Nutritional status, symptom burden, and predictive validity of the Pt-Global web tool/PG-SGA in CKD patients: a hospital based cross sectional study.

Dear Dr. Tiwari,

Thank you for submitting your manuscript to PLOS Global Public Health. After careful consideration, we feel that it has merit but does not fully meet PLOS Global Public Health’s publication criteria as it currently stands. Therefore, we invite you to submit a revised version of the manuscript that addresses the points raised during the review process.

We look forward to receiving your revised manuscript.

Kind regards,

Ransome Eke, M.D., Ph.D., MPH

Academic Editor

Journal Requirements:

Additional Editor Comments (if provided):

Reviewers' comments:

Reviewer's Responses to Questions

**Comments to the Author**

1. If the authors have adequately addressed your comments raised in a previous round of review and you feel that this manuscript is now acceptable for publication, you may indicate that here to bypass the “Comments to the Author” section, enter your conflict of interest statement in the “Confidential to Editor” section, and submit your "Accept" recommendation.

Reviewer #1: All comments have been addressed

Reviewer #2: (No Response)

2. Does this manuscript meet PLOS Global Public Health’s publication criteria? Is the manuscript technically sound, and do the data support the conclusions? The manuscript must describe methodologically and ethically rigorous research with conclusions that are appropriately drawn based on the data presented.

Reviewer #1: Yes

Reviewer #2: Partly

3. Has the statistical analysis been performed appropriately and rigorously?

Reviewer #1: Yes

Reviewer #2: Yes

4. Have the authors made all data underlying the findings in their manuscript fully available (please refer to the Data Availability Statement at the start of the manuscript PDF file)?

Reviewer #1: Yes

Reviewer #2: Yes

5. Is the manuscript presented in an intelligible fashion and written in standard English?

Reviewer #1: Yes

Reviewer #2: No

6. Review Comments to the Author

Reviewer #1: I have no further comment. Good luck!

Reviewer #2: 1. The study is not well organized; not understandable for the readers.

2. The objectives of the study are not well settled.

3. The study has many outcomes/objectives as indicated indifferent parts/segments of it

4. The gap of the study is also not well dressed and not well focused

5. Study design is not clear/lack of consistency and appropriateness

6. Conclusion and Recommendation should be done on the study finds only in the precise form.

7. Same of the references are not update

8. Please edit your work (language)

7. PLOS authors have the option to publish the peer review history of their article (what does this mean?). If published, this will include your full peer review and any attached files.

**Do you want your identity to be public for this peer review?** For information about this choice, including consent withdrawal, please see our Privacy Policy.

Reviewer #1: **Yes: **LEUNG Yuen Ling

Reviewer #2: No

---

## [Editor Report · Decision Letter 2]

19 Dec 2022

Nutritional status, symptom burden, and predictive validity of the Pt-Global web tool/PG-SGA in CKD patients: a hospital based cross sectional study.

PGPH-D-22-01367R2

Dear Dr. Tiwari,

We are pleased to inform you that your manuscript 'Nutritional status, symptom burden, and predictive validity of the Pt-Global web tool/PG-SGA in CKD patients: a hospital based cross sectional study.' has been provisionally accepted for publication in PLOS Global Public Health.

Best regards,

Ransome Eke, M.D., Ph.D., MPH

Academic Editor